# Efficacy and safety of albendazole 400 mg for 30 days compared to single dose of ivermectin in adult patients with low *Loa loa* microfilaremia: A non-inferiority randomized controlled trial

Luccheri Ndong Akomezoghe, Noé Patrick M'Bondoukwé ID *,
Denise Patricia Mawili Mboumba, Jacques Mari Ndong Ngomo,
Bridy Chesly Moutombi Ditombi, Coella Joyce Mihindou,
Roger Hadry Sibi Matotou, Valentin Migueba, Marielle Karine Bouyou Akotet

Department of Parasitology-Mycology-Tropical Medicine, Faculty of Medicine, Université des Sciences de la Santé, Libreville, Gabon

\* mbondoukwenoe@gmail.com

## Abstract

### Background

*Loa loa* infection is endemic in central African countries and particularly in Gabon. Treatment typically involves the use of ivermectin and albendazole, with albendazole often administered to reduce microfilaremia in individuals with high microfilaremia before taking ivermectin. This study aims to evaluate the efficacy and safety of albendazole in patients with low microfilaremia.

### Methodology and principal findings

The study was conducted from November 2021 to April, 2022 in the Woleu-Ntem province of northern Gabon. Following a questionnaire, direct examination of 10 μL of blood and leukoconcentration technique were perfomed for *Loa loa* detection. Of 406 identified microfilaremic cases, 48 volunteers were randomized, 21 women and 27 men, their mean age was $51 \pm 13$ years. Overall, 24 received, daily 400 mg albendazole for 30 days and 24 others were treated with a single course of 200 μg/kg ivermectin. Microfilaremia and adverse events were monitored from D0 to D30. In the per-protocol analysis, the mean microfilaremia decreased significantly by 82.3% and 90.4% in the albendazole and ivermectin groups, respectively ($p< 0.001$). The risk difference between both treatments was 8.1% [95% CI: 16.8; -0.6%]. In the intention-to-treat analysis, the mean microfilaremia decreased significantly by 82.4% and 90.8% in the ALB and IVM groups, respectively ($p< 0.001$), with a risk difference of 8.4% [95% CI: 16.2; 0.6%]. Eosinophil levels decreased by day 30, although they were not significantly different following albendazole and ivermectin treatments.

**Data availability statement:** The authors confirm that all data underlying the findings are fully available without restriction. All relevant data are within the paper and its Supporting Information files.

**Funding:** This project is part of a larger programme funded by the European and Developing Countries Clinical Trials Partnership (EDCTP) under the TMA (Training and Mobility Actions) 2019 Career Development Fellowship (CDF) – Grant No. TMA2019CDF-2730, focusing on the treatment of hypermicrofilaraemic loiasis and the evaluation of various albendazole protocols. The fellowship was awarded to NPM. All patients with low microfilaraemia ranging from ≥500 to ≤3500 mf/mL were included in the present study. However, no funding was allocated for the purchase of ivermectin or to cover the publication costs of the study results. The funders had no role in the study design, data collection and analysis, decision to publish, or preparation of the manuscript.

**Competing interests:** The authors have declared that no competing interests exist.

## Conclusions/Significance

Albendazole demonstrated microfilaricidal activity in individuals with low *Loa loa* microfilaremia following a 30-day treatment. The monitoring of parasite density 3–10 months post-treatment is needed to complete the present findings.

### Author summary

*Loa loa* is a parasitic worm transmitted by *Chrysops* flies, endemic in the region of Central Africa, notably in Gabon. High levels of microfilaremia (> 8,000 mf/mL) can cause serious adverse effects when treated with ivermectin, a drug widely used in mass treatment campaigns. For this reason, albendazole is sometimes administered beforehand to reduce the parasite load in at-risk individuals. However, few data exist concerning its efficacy in people with low microfilaraemia, even though albendazole is the most accessible drug for rural populations, and the majority of patients with loiasis have low microfilaraemia levels below 4,000 mf/mL. In this study, we compared the efficacy and tolerability of albendazole (400 mg/day for 30 days) *versus* a single dose of ivermectin (200 µg/kg) in low-microfilaremic individuals in northern Gabon. Both treatments led to a significant reduction in microfilaremia, with a slightly higher efficacy observed for ivermectin. No serious adverse events were reported in either group. These results support the potential use of albendazole as an alternative treatment in loiasis endemic areas, particularly when this treatment is more accessible than ivermectin for the populations concerned.

## Introduction

Loiasis is a neglected filarial parasitic disease endemic to West and Central Africa [1]. It is caused by the *Loa* (*L.*) *loa* worm, with adult worms residing in the skin or intermuscular fascia and microfilariae found in the blood. Over 10 million people are estimated to be carriers of this parasite [2]. Common clinical manifestations of loiasis include Calabar swelling, eyeworm, subconjunctival migration, subcutaneous crawling, and pruritus [3,4]. There is no specific drug developed exclusively for *L. loa* treatment; current treatment strategies primarily rely on the use of diethylcarbamazine (DEC), ivermectin (IVM), and albendazole (ALB).

The World Health Organization (WHO) recommends the deployment of mass drug administration (MDA) with IVM to eradicate onchocerciasis and lymphatic filariasis [5,6]. In regions where onchocerciasis coexists with loiasis, MDA with IVM could be responsible for serious post-treatment reactions in individuals with *L. loa* hypermicrofilaremia, which is defined for parasite densities higher than 8,000 microfilariae per milliliter of blood (mf/mL). Indeed, patients with this condition may experience fatal and/or serious adverse events (SAEs) following IVM treatment, making areas with onchocerciasis-loiasis co-endemicity ineligible for MDA with IVM [7,8]. In areas such

as Gabon, where onchocerciasis is hypoendemic, a Test and Treat strategy is considered as an alternative [9–11]. Several studies have been conducted to evaluate treatments that could safely reduce high *L loa* microfilaremia, thereby making populations eligible for IVM treatment. Different albendazole (ALB) regimens were tested using various regimens. In Cameroon, administering ALB 400 mg for three days showed a reduction in microfilaremia at day 90 [12]. When patients with high *L loa* microfilaremia were given two or six doses of ALB 800 mg every two months, a decrease in microfilaremia was observed in patients receiving six doses after four months [13].

Recently in Gabon, a five week regimen of ALB 400 mg showed a reduction in microfilaremia comparable to a treatment with ALB 400 mg for three weeks coupled with IVM [14]. On one hand, these studies highlight the microfilaricidal effect of ALB; on the other hand, they underscore the lack of a standardized protocol for treating microfilaremic loiasis.

Although initially considered benign, today, numerous studies suggest that loiasis should be regarded as a major health issue and added to the list of neglected tropical diseases [15,16]. Its prevalence can reach 50% in some areas of Gabon which is considered as hyperendemic for this filariasis [3,11,17,18]. Most loiasis patients have low microfilaremia (<4,000 mf/mL) and reside in rural areas [3,17,18]. Indeed, local populations primarily engage in agriculture, fishing and hunting, which generally do not provide substantial income. While clinically, low microfilaremia or occult loiasis is often associated with frequent subjective and objective symptoms such as Calabar swelling, due to this non negligible morbidity, patients often seek care in traditional medicine [19]. For those who have access to local dispensaries, their main health facilities, albendazole is the only available and affordable medication. Since more than 20 years, onchocerciasis control program activities through IVM mass drug administration are not performed in Gabon. Therefore, IVM is not available in most of the rural provinces and the treatment cost is more than 20 USD. Albendazole was shown to be safe, without serious adverse events for the treatment of *L. loa* high microfilaremia (> 30,000 mf/mL) [13,14]. Moreover, a one month treatment regimen costs less than 5 USD, which is the price for fifty 400 mg tablets (Vendor at national level: UbiPharm Gabon, Reference: Albendazole ubigen 400 mg 50 blisters of 1 tablet, Manufactured by: Lincoln Pharmaceutical Ltd for Ubithera Pharma PVT. Ltd). Thus, it would be a safe and affordable treatment option for low *L. loa* microfilaremia.

Therefore, the aim of this study was to evaluate the efficacy and safety of a 400 mg daily dose of ALB for 30 days compared to a single dose of IVM for treating infected patients with low microfilaremia in Gabon.

## Materials and methods

### Ethics statement

This study was a part of the PHYLECOG project funded by the EDCTP2 program under reference TMA2019CDF-2730. The study protocol was reviewed and approved by the National Ethics Committee for Scientific Research of Gabon. Written informed consent was obtained from each participant after explaining the study (Protocol No. 0053/2022/CNER/P/SG). The study is registered under ISRCTN14889921.

### Study sites and population

This survey was conducted among a rural population of Woleu-Ntem, specifically in the departments of Ntem and Haut Ntem. Woleu-Ntem is a province located in the northern region of Gabon, a Central African country. The province has an equatorial climate characterized by hot, humid conditions and high rainfall. It is covered with old secondary forests and has rich and varied wildlife [20]. In Woleu-Ntem, the overall prevalence of *L. loa* is 20.2% [18]. Rural populations live along roads and rivers, with farming being their primary activity. Fishing and hunting are also common, which increases their exposure to Chrysops bites, the vectors responsible for transmitting *L. loa* [18].

### Study design

This study was a single blinded non-inferiority randomized controlled trial aimed at evaluating the efficacy and safety of a 400 mg dose of ALB taken daily for 30 days versus a single dose of IVM at 200 µg/kg. The study was conducted from

November 2021 to April 2022, in small villages within the province of Woleu-Ntem, located in northern Gabon. A preliminary visit was made to inform local authorities about the study's purpose. After obtaining informed consent, participants' sociodemographic and clinical data were collected. This included information on the study site, participant identification, age, sex, weight, height, and clinical symptoms related to loiasis, such as worm migration into the eye, Calabar swelling, adult worm migration under the skin, and pruritus.

### Inclusion and non inclusion criteria

Eligible participants were individuals aged 18–75 years who were diagnosed with *L. loa* infection confirmed by microscopy, with microfilarial counts ranging from ≥500 to ≤3500 mf/mL.

Those who reported a benzimidazole treatment during the three previous months, who had known acute systemic illnesses or suspected allergy to benzimidazoles, or pregnant or lactating women were not included.

### Withdrawal criteria

Volunteers were withdrawn from the clinical trial if a reduction in *L. loa* microfilaremia was not observed at D14, or if there was a significant worsening of the clinical symptoms.

### Sample size calculation

Sample size was determined using the formulation of sample size calculation for non-inferiority trials [21].

$$N = \frac{(pA * (1 - pA) + pB * (1 - pB)) * ((qnorm(1 - alpha)) + qnorm(1 - beta))}{(pA - pB - delta)\verb|^|2}$$

Thus, based on an expected 73% (pB) reduction in microfilaremia with ALB and a 90% (pA) reduction with IVM [22–24], and assuming a first-species risk of 5.0% (alpha) and a second-species risk of 10.0% (beta), with a non-inferiority margin of 20% (delta), it was determined that a minimum of 18 (N) patients per group would be required for this study.

### Blood collection and microfilaremic loiasis diagnosis

Venous blood was collected in an EDTA tube for parasitological diagnosis. Overall, 10 μL of fresh whole blood were used for the direct examination and 4 ml for the leukoconcentration technique as described by Ho Thi Sang and Petithory [25]. The microfilaremia was expressed as the number of mf/mL. Absence of microfilaremia was defined in the absence of detectable parasites following both microscopic techniques.

### Randomization and treatment procedure

Individuals were randomly assigned to one of two treatment groups following stratification based on the level of microfilaremia. The ratio of allocation within the two groups was 1:1 based on three levels of microfilaremia: 500–1500; 1501–2500; 2501–3500 mf/mL. A randomization list was prepared by a statistician using a random digit table and different written numbers were provided to the study investigators, who were unaware of the contents of the randomization list. The randomization list was provided to an independant pharmacist who was not involved either in patient selection or inclusion, or in patient folllow-up, data collection, or analysis. For each patient, a number was randomly drawn from the pool of digit provided by the statistician. The pharmacist assigned albendazole or ivermectin treatment to the volunteer based on its corresponding position on the list of randomization. Treatment was administered as follows: patients either received a daily dose of one tablet of ALB (400 mg) for 30 days or a single calculated dose of IVM (200 μg/kg) based on the patient's weight. Both treatments were supplemented with 3 mg of antihistaminic-H2 daily for 7 days. Other infected participants who where not randomized were treated according to the reference center protocol. To ensure the single-blinded nature

of the trial, participants were not informed of their treatment group or the duration of their regimen. Daily interactions with study personnel were maintained in both groups, and all participants received similar clinical follow-up and antihistamine treatment during the first week. Although the treatment durations differed, these measures helped minimize the participants' awareness of group allocation.

### Monitoring of clinical signs and adverse events

Symptoms and signs attributable to loiasis were recorded before and throughout the study. In addition, adverse events that corresponded to new symptoms or worsening of existing ones were systematically evaluated throughout the study and categorized by severity: mild if they caused minimal discomfort and did not disrupt daily activities, moderate if they interfered with daily activities to a noticeable extent, and severe if they significantly impaired the patient's ability to carry out normal daily tasks.

### Efficacy assessments

Blood samples for direct examination (10 μL) and leukoconcentration were collected at baseline, prior to administration of the study drug, and on days 2, 7, 14, and 30 to assess microfilarial load and eosinophilia throughout the follow-up period. Given the diurnal periodicity of the filarial species, all efforts were made to collect blood samples between 10:00 am and 3:00 pm.

### Monitoring of eosinophil count

Eosinophils were counted from a thin blood smear prepared with 5 μL of fresh whole blood sample. The eosinophil count was expressed in percentage, based on 100 white blood cells observed under a light microscope. A count of ≥ 7% was considered indicative of hypereosinophilia. Blood smears were conducted at baseline, prior to administration of the study drug, and on days 2, 7, 14, and 30 to assess the eosinophil levels.

### Outcomes/endpoints

The primary endpoint of this study was the reduction rate of microfilaremia. Non-inferiority between a monthly course of ALB and a single course of IVM was assessed based on the difference in reduction rates at the last time point, on day 30.

Treatment with ivermectin for microfilaremic loiasis typically results in rapid and sustained reduction, achieving a reduction rate of approximately 90% after 30 days, whereas placebo generally achieves a 50% reduction [22,24].

To establish the non-inferiority, a margin of 20% was selected using the fixed margin method, which is half the difference in reduction rates observed between ivermectin and placebo.

The secondary endpoint was the disappearance of suggestive symptoms of loiasis.

Adverse events were monitored and stratified by type and severity. An adverse event was any untoward medical event related or unrelated to the study medication, which appeared after the first administration of the drug,. The primary safety endpoint was the occurrence of any adverse event following treatment initiation, regardless of its relationship with the study drug. The secondary safety endpoint involved the evolution of eosinophil levels from D0 to D30.

### Statistical analysis

Data processing was conducted using Microsoft Excel 2016, overseen by both an operator and a checker. R software version 4.3.0 was used for comprehensive data analysis.

The statistical analyses were performed according to CONSORT 2010 guidelines for per-protocol and intention-to-treat methodologies [25]. The per-protocol analysis focused exclusively on patients who completed their prescribed treatment regimen and all the follow-up visits. Conversely, the intention-to-treat analysis included all randomized patients, irrespective of the number of follow-up visits attended.

Categorical data such as sex, symptoms, and adverse events were summarized as absolute values and percentages. These were compared using the proportion test, given the sample size was less than thirty, which is suitable for such analyses.

Continuous variables were evaluated for normality. Variables following a normal distribution, such as age, weight, height, BMI, and microfilaremia, were presented as geometric mean or arithmetic mean ± standard deviation. Variables that did not follow a normal distribution, like eosinophil level, were presented as medians [25th percentile – 75th percentile].

The reduction rate of microfilaremia was calculated as a percentage using the formula: ((Microfilaremia at D0 -Microfilaremia during follow-up)/ Microfilaremia at D0) * 100. According to the CONSORT guideline for non-inferiority studies, the risk difference of the main outcome between the control group (IVM treatment) and the investigational product (ALB) group was determined [26]. It was calculated in per protocol and Intention to Treat (ITT) analysis with its respective 95% confidence interval. An exploratory analysis was performed using a Chi2 test, comparing the observed reduction rates in the per-protocol and the intention-to-treat analysis.

The comparisons of microfilaremia or eosinophil levels were performed between Day 0 and Day 30 values using Student's t test for normally distributed variables or the Wilcoxon signed rank test for variables with non-normal distribution. Data off other time points were used to illustrate the evolution of these variables during the participant follow-up. A $p$-value inferior to 5% was considered significant for all analysis.

## Results

### Patients

Among the 1342 volunteers who underwent blood testing, 406 (30.2%) had *L. loa* microfilaremia. Finally, 48 patients were enrolled, with 24 treated with ALB and 24 with IVM. During the follow-up period, 10 patients withdrew their consent (Fig 1).

### Demographic and baseline characteristics

The characteristics of the study population are summarized in Table 1. There were no significant differences between the two groups of patients according to gender, age, weight, height, microfilaremia, and eosinophil level.

### Efficacy of treatments on the reduction of microfilaremia

**Per-protocol analysis.** Both treatment groups showed significant reductions in mean microfilaremia from day 0 to day 30. Specifically, the ALB group exhibited a reduction of 82.3% (1339 to 237 mf/mL), while the IVM group showed a reduction of 90.4% (1395–134 mf/mL) (p < 0.01) (Fig 2). The microfilaremia reduction in the ALB group was significantly higher than the reduction in the IVM group (p < 0.001). The risk difference between the two regimens was 8.1% [95% CI: 16.8; -0.6%], and the 95% confidence interval did not exceed the margin of non-inferiority.

**Intention-to-treat analysis.** The mean microfilaremia in patients from both treatment groups decreased significantly by 82.4% (1350–237 mf/mL) and 90.8% (1454–134 mf/mL) from day 0 to day 30 in the ALB and IVM groups, respectively (p < 0.001) (Fig 3). The reduction in microfilaremia differed significantly between the treatments (p < 0.001). The risk difference between the two regimens was 8.4% [95% CI: 16.5; 0.7%], and the 95% confidence interval did not exceed the margin of non-inferiority.

### Secondary outcomes

**Incidence and evolution of loiasis-attributable signs during treatment.** At baseline, approximately half of the patients in both groups reported suggestive symptoms of loiasis: 41.7% (n = 10/24) in the ALB group and 54.2% (n = 13/24) in the IVM group (p = 0.6). The most frequent symptom was pruritus, noted in 90.0% (n = 9/10) of patients in the ALB

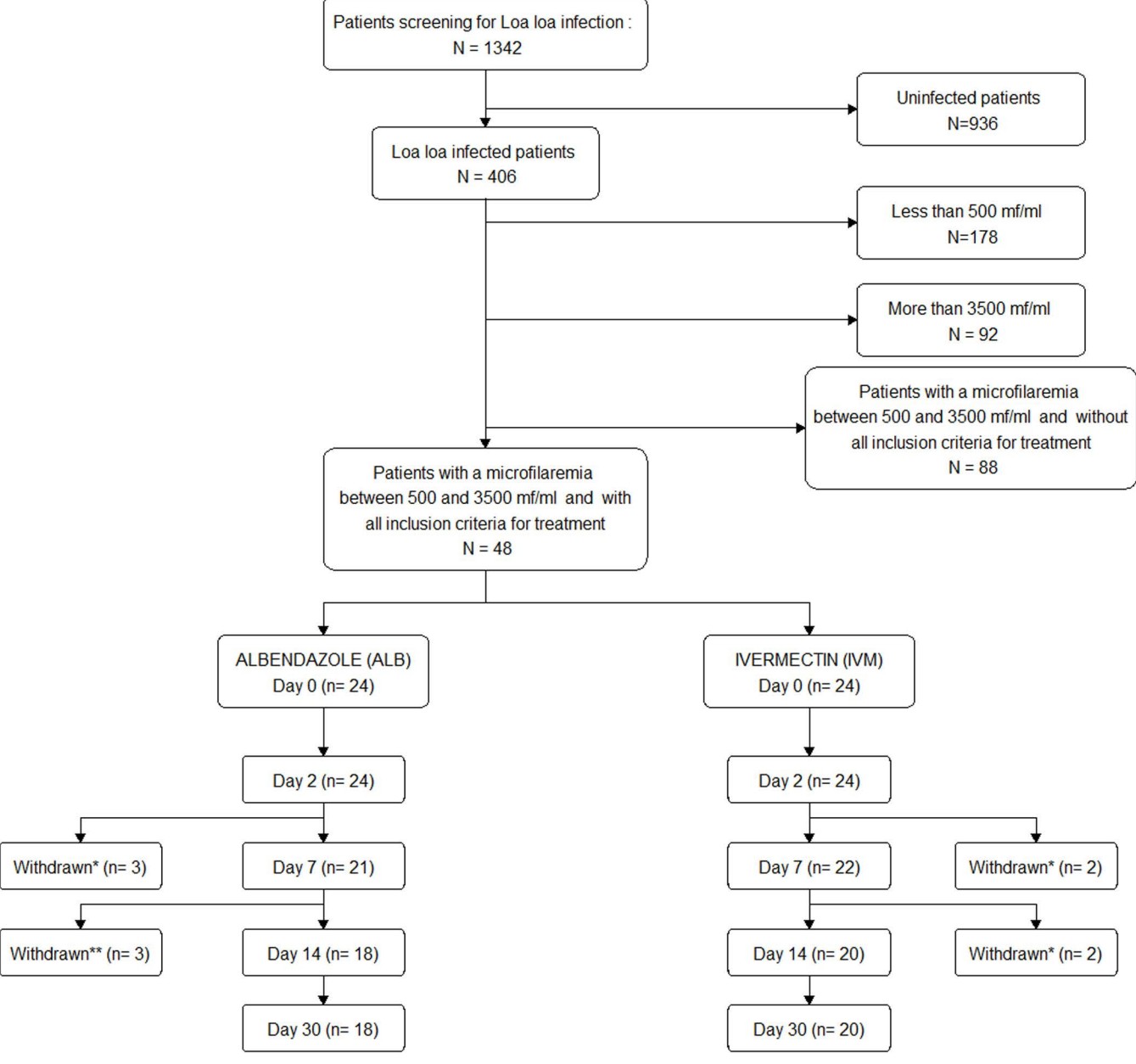

**Fig 1. Study flowchart.** * Refusal of venipuncture; ** Long-term traveler; did not attend the follow-up visit and could not be sampled at the appropriate times for microfilaremia assessment, but present at the following visit.

group and 69.2% (n = 9/13) in the IVM group. During the follow-up period, the number of patients experiencing symptoms decreased, with no patients reporting symptoms by day 30 in either treatment group ([Fig 4](Fig 4)).

**Incidence and evolution of adverse events during treatment.** Globally, a total of 10 on 24 patients (41.7%) in the ALB group and 8 on 24 patients (33.3%) in the IVM group reported adverse events by day 2 (p = 0.8). The most common adverse event in the ALB group was asthenia, reported by 80.0% (n = 8/10) of those experiencing adverse events. In the IVM group, the most common adverse events were asthenia, reported by 50.0% (n = 4/8), and headache, reported by 25.0% (n = 2/8) ([Fig 5](Fig 5)).

**Table 1. Descriptive characteristics of the two treatment groups at the inclusion (D0).**

| Variables | ALBENDAZOLE (N = 22) | IVERMECTIN (N = 24) | P value |
|---|---|---|---|
| **Male, n (%)** | 17 (70.8) | 10 (41.7) | 0.08 |
| **Age in years,** mean ± SD | 49.8 ± 12.5 | 53.0 ± 13.6 | 0.4 |
| **Weight in Kgs,** mean ± SD | 64.3 ± 10.3 | 64.5 ± 10.7 | 0.9 |
| **Height in meters,** mean ± SD | 1.6 ± 0.1 | 1.6 ± 0.1 | 0.2 |
| **BMI in Kg/m²,** mean ± SD | 23.6 ± 3.2 | 24.6 ± 4.2 | 0.4 |
| **Microfilaremia in mf/mL,** median [IQR] | 1150 [700–1625] | 1350 [875–2000] | 0.5 |
| **Eosinophil rate, in %,** median [IQR] | 16.0 [10.5–21.0] | 16.0 [12.5–25.5] | 0.5 |

SD: standard deviation, [IQR]: [25th percentile – 75th percentile].

**Efficacy of treatments on eosinophil level reduction.** Eosinophil levels decreased by Day 30, although the difference from Day 0 to Day 30 was not statistically significant in the ALB (p = 0.99) or IVM groups (p = 0.18). In the ALB group, eosinophil levels initially increased from Day 0 to Day 2, followed by a decline across subsequent visits, reducing from a median rate of 16.0% to 7.0% (Fig 6). Conversely, in the ivermectin group, eosinophil levels remained stable from Day 0 to Day 14, with a decrease observed by Day 30 from a median level of 16% to 4% (Fig 6).

## Discussion

This clinical trial represents the first evaluation of a 30-day ALB treatment for the reduction of low *L. loa* microfilaremia, a common feature in continuously exposed individuals. The ALB treatment regimen was not inferior to a single course of IVM for the reduction of *L. loa* microfilaremia under the threshold of 500 mf/mL, as assessed one month after the first dose administration.

This study contributes to evidence-based data supporting tailored loiasis treatment recommendations in highly endemic areas where no consensus has existed until now. Low levels of microfilaremia is common in endemic Central Africa settings. In the Republic of Congo, in Cameroon, and in other studies performed in Gabon, the median microfilaremia rarely reached 1500 mf/mL [17,18,26–28]. The data presented here highlights the high frequency (77.3%; n= 314/406) of low *Loa loa* microfilaremia carriers in a large population (more than 1000 participants) as also reported by Akue 10 years ago, and recently by Veletzky et *al.* [29].

Trials performed in endemic or non endemic areas included either patients with hypermicrofilaremia, short treatment duration, or sequential treatment administration [13,30,31]. Short or sequential treatment courses, although with high dosages (400 or 800 mg ALB daily) were not associated with a significant reduction of parasitaemia, even 12 to 24 months following the start of the treatment [13,31]. Indeed, it is suggested that a short course of ALB does not significantly reduce microfilaremia and that a longer treatment course with higher dosages likely has more significantly embryotoxic and adulticidal effects [31,32]. Gobbi et *al.* treated *L. loa* infected patients with 400 mg daily for 28 days, followed by a single dose of IVM, resulting in blood clearance of microfilaremia in 93.0% of treated participants within 6 months [33]. In the region of Lambarene, a shorter course of ALB 800 mg daily for 21 days followed by a single dose of IVM, or a longer course (35 days) of 800 mg ALB, resulted in a significant parasite reduction rate at day 30 (91.0 to 96.0%) [14]. A retrospective analysis of different regimens of ALB treatment according to the level of microfilaremia, during the routine management of patients seen at the Department of Parasitology-Mycology-Tropical Medicine (DPMTM), showed a significant reduction or complete cure rates between 6 to 12 months. The present trial was designed to assess the efficacy of one of these home-based treatment regimens (a course of 400 mg ALB daily for 30 days) following standard procedures of a randomized efficacy trial, thus to provide scientifically evidence-based data.

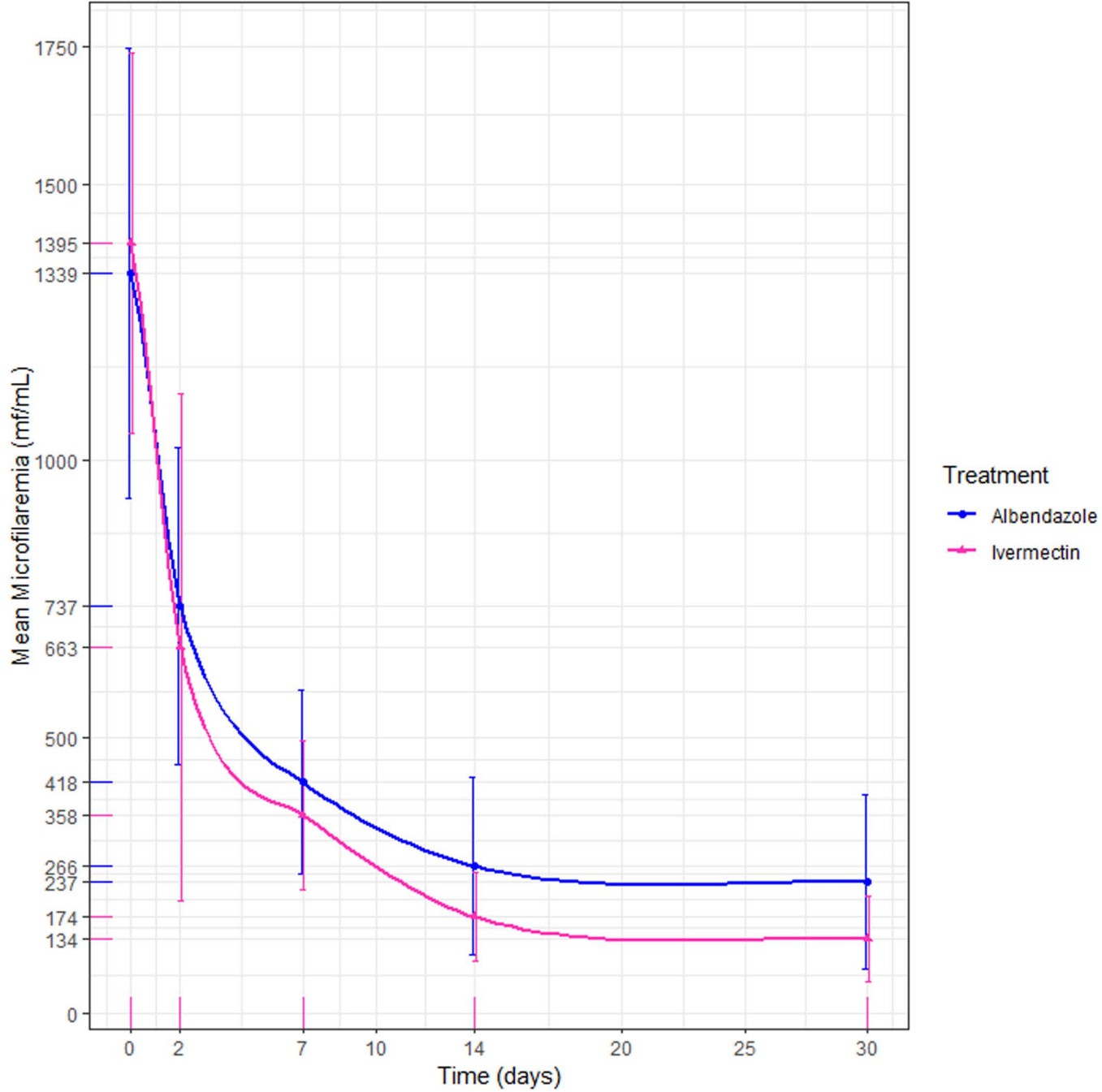

**Fig 2. Comparison of the albendazole and ivermectin parasitological cure rate (Per-protocol).** The blue line on the chart represents the microfilaremia levels of patients treated with albendazole, while the pink line represents those treated with ivermectin.

In both the intention-to-treat (8.6%) and the per protocol (8.1%) analysis, the ALB and IVM treated participants experienced a significant decrease in mean microfilaremia from day 0 to day 30. The 95% confidence interval for these differences observed did not reach or exceed the non-inferiority margin of 20%, according to the CONSORT guidelines, thus confirming the non-inferiority of a month-long course of ALB compared to single-dose IVM.

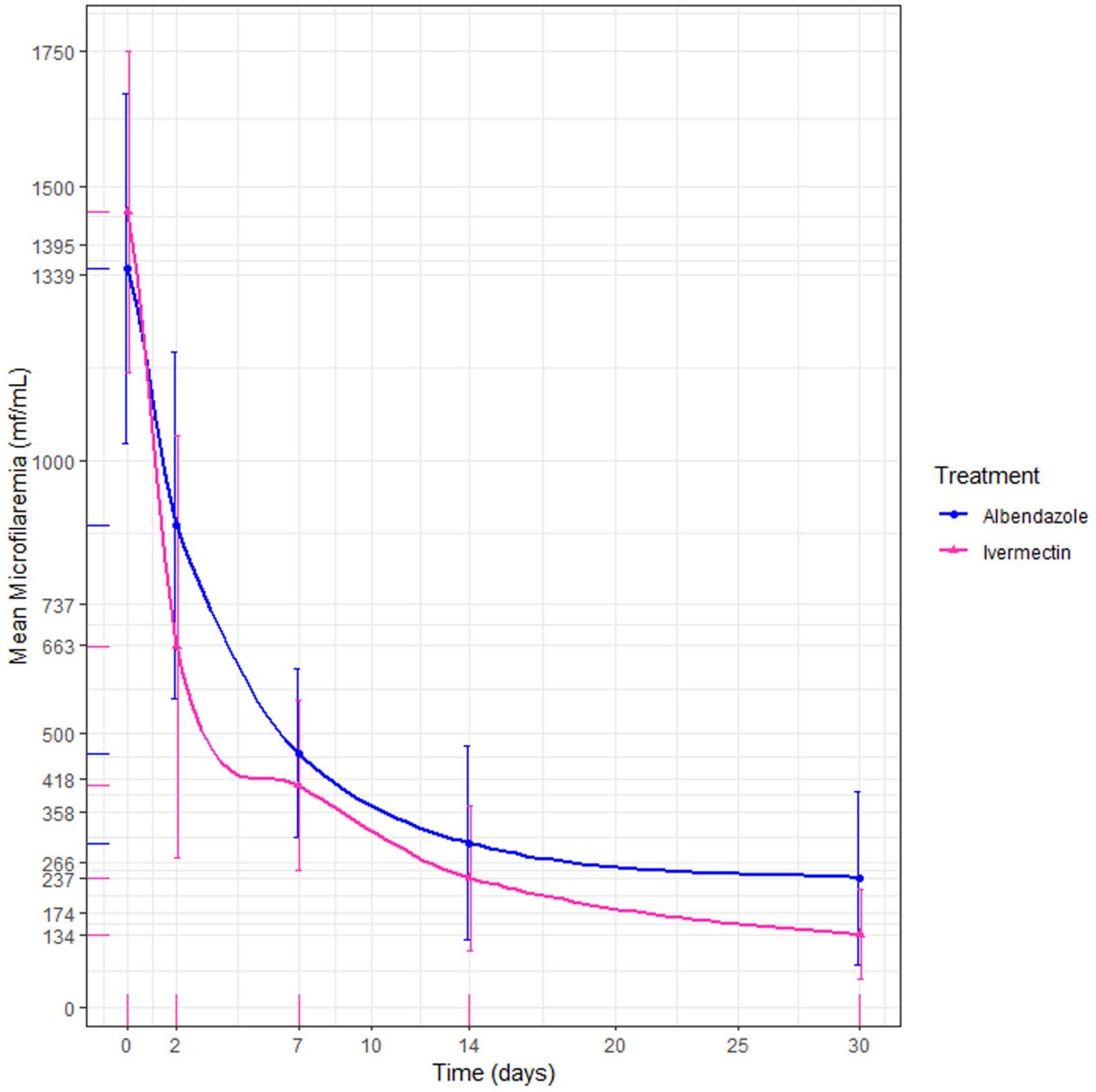

**Fig 3. Comparison of the albendazole and ivermectin parasitological cure rate (Intention-to-treat).** In this line chart, the blue line represents the microfilaremia levels of patients treated with albendazole, while the pink line represents those treated with ivermectin.

In other trials, the observed reduction of *L. loa* microfilaremia was highly significant at Day 30-35 as recorded among our study participants [14,33]. It is interesting to highlight that in hypermicrofilaremic patients, even after 4 to 6 months following a 35 days of albendazole treatment regimen, there was no significant resurgence of high microfilaremia, but, a higher proportion of cured participants in months 6 to 12 [14]. Thus, we can hypothesize that, in cases of low

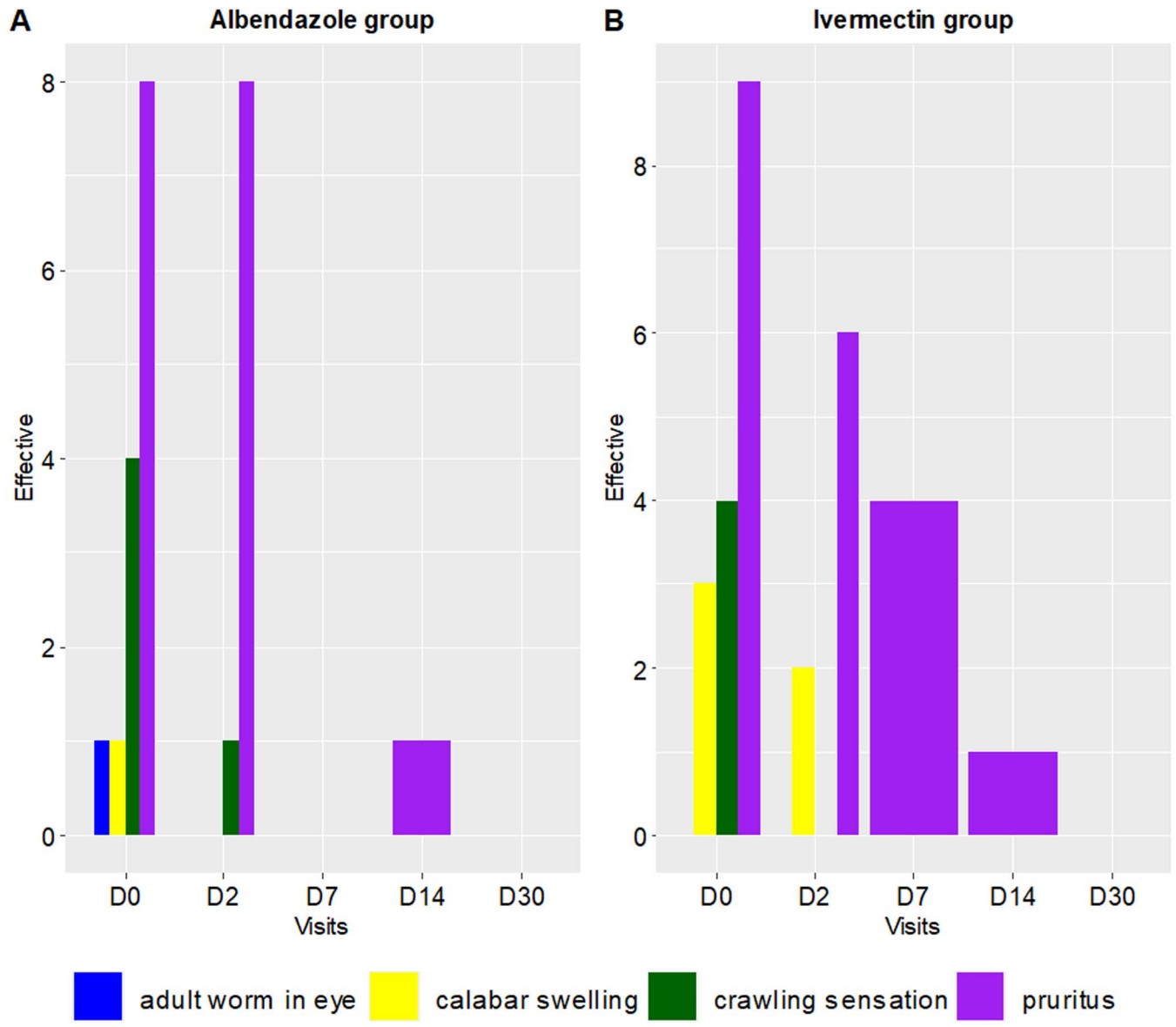

**Fig 4. Evolution of the frequency of *Loa loa* clinical symptoms over the 30-day follow-up period.** The histogram shows adult worms in the eye with blue bars, Calabar swelling with yellow bars, crawling sensation with green bars, and pruritus with purple bars.

microfilaremia, an acceptable cure or parasite clearance rate would be achieved with our proposed regimen after 6 to 12 months without the need of an additional treatment. Clinical trials with larger sample sizes and longer follow-up as well as *in vitro* evaluations are needed to confirm this hypothesis.

During follow-up, the number of patients with symptoms decreased, and no patient exhibited any loiasis symptoms by day 30 in either treatment group. This highlights an adequate clinical response to both treatments. However, these findings contrast with the results of the retrospective observational study by Gobbi *et al*., where 31.3% of patients treated with a single dose of IVM and 50.0% of those treated with ALB (400 mg/day) were still symptomatic after 30 days of treatment [33].

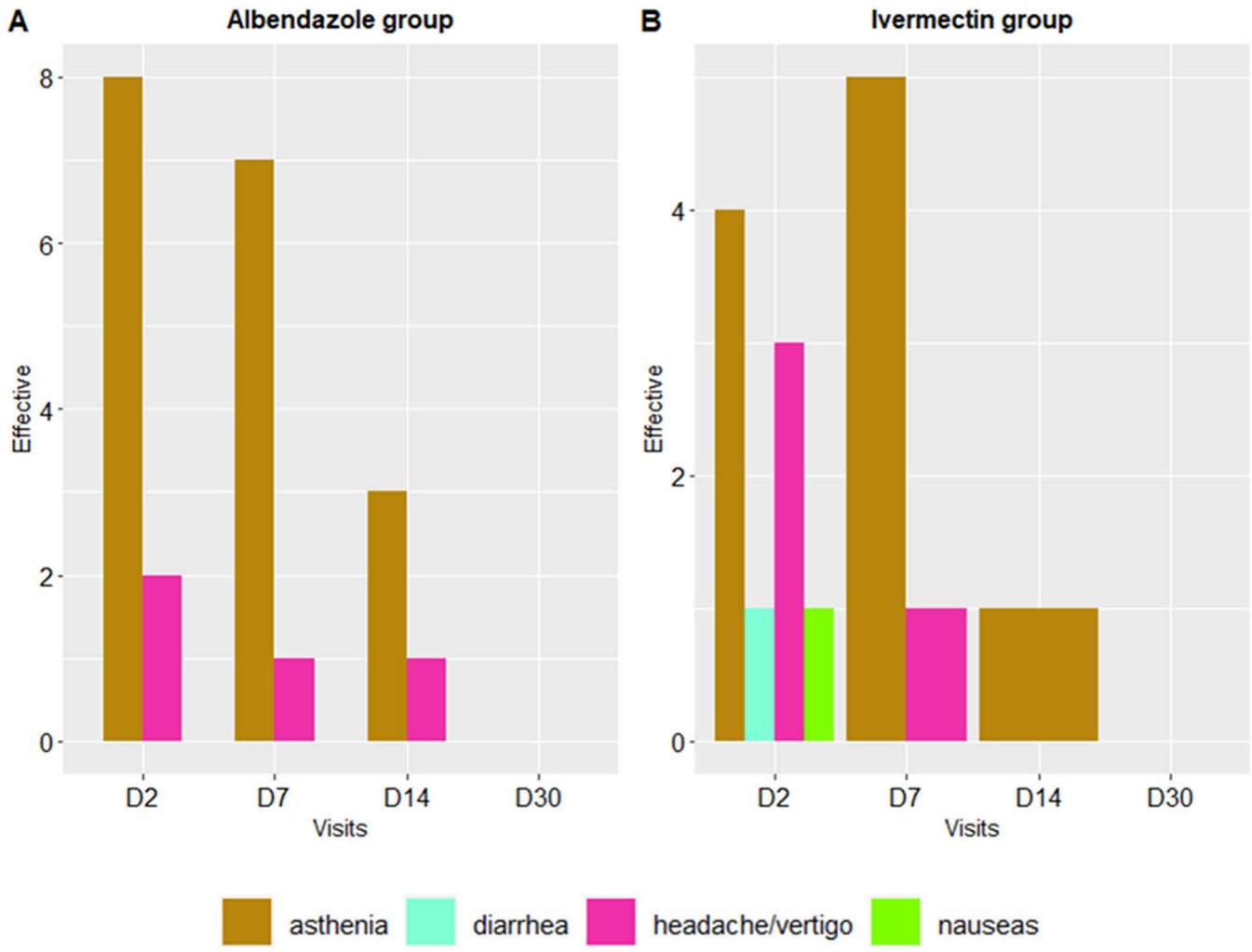

**Fig 5. Evolution of clinical adverse events over the 30-day follow-up period.** The histogram shows appetite disorders (pink bars), asthenia (gold bars), diarrhea (aquamarine bars), headache and/or vertigo (fuchsia bars), and nausea (green bar).

Less than half of the patients in both treatment groups experienced adverse events during the follow-up period, with asthenia being the most common symptom in both groups. The frequency of these adverse events decreased over subsequent visits, indicating the treatments' safety, all of which were mild. Similar findings have been reported in prior studies using comparable dosages, where no participants experienced serious adverse events [33,34]. Regarding the impact of both treatments on eosinophil counts, there was no statistically significant reduction observed in either treatment group during the follow-up period. These results are consistent with Herrick et al., who found no decrease in eosinophil counts after two weeks of IVM treatment [35]. They also align with Klion et al., who reported no reduction in eosinophil counts after 21 days of ALB treatment [23]. Globally, a significant decrease of eosinophil count is usually observed between 6 and 24 months post-treatment [36,37].

This study has some limitations. First, the sample size was small according to the high endemicity of loiasis in Gabon, and additional data with larger populations need to be generated. Secondly, a 30-day treatment with 400 mg ALB per day could be associated with various compliance problems, notably due to the prolonged duration of

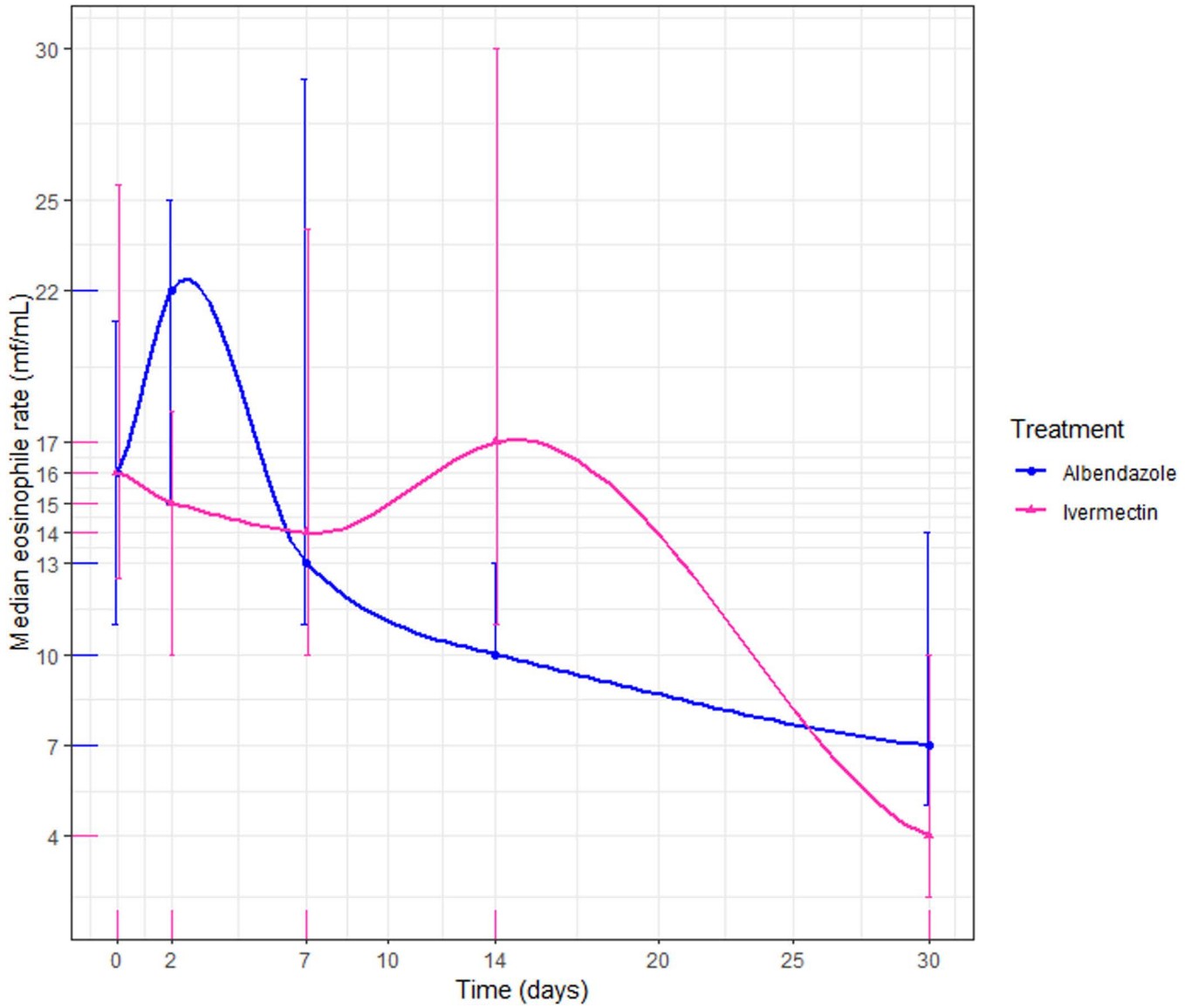

**Fig 6. Eosinophil rate progression by treatment.** The blue line on the chart depicts the eosinophil rate of patients treated with albendazole, while the pink line represents those treated with ivermectin.

treatment, which could lead to reduced adherence. While no adherence observance evaluation was performed during our routine practice in the DPMTM, patients, who are usually followed up every 3 months for 6 to 12 months, do not usually complain (data from the DPMTM). Thirdly, ALB semi-annual single dose administration is suspected to be associated with a decrease in sensitivity of *Trichuris trichiura* to this drug [38]. Nevertheless, there are no data on the impact of long treatment courses of ALB. Such treatment would cure intestinal helminth infections and would be associated with a significant reduction of Soil-Transmitted Helminthiasis (STH) transmission in areas with high *L. loa* and STH co-endemicity such as rural settlements of Gabon [17,39]. Finally, the relatively short observation period of 30 days allows to well characterizing the initial response to the treatment but does not allow to obtain information about the long-term results.

Strengths of the study include its randomized design, prospective enrollment, and comprehensive statistical analysis using both per-protocol and intention-to-treat methods. Furthermore, the present study results suggest that a month-long course of daily 400 mg ALB has acceptable parasitological and clinical efficacy, and is well tolerated and safe for the reduction of low microfilaremia. Although a longer follow-up duration is needed, these data are valuable information that would contribute to the development of guidelines and recommendations for the treatment of continuously exposed *Loa loa* infected individuals.

## Conclusion

A 30-day ALB treatment demonstrated an adequate clinical response and good safety profile that was non-inferior to a single dose of IVM for the reduction of low *L. loa* microfilaremia in the study population. Additional studies with larger sample sizes and extended follow-up periods should be performed to confirm the efficacy and safety of this ALB regimen as an efficacious, safe and affordable treatment option for the case management of loiasis of low microfilaremia in areas without easy access to IVM.

## Supporting information

**S1 Table. Dataset.** The dataset of the provided results.
(PDF)

**S2 Table. Data dictionary.** The table defined all variables which are in the dataset of S table.
(PDF)

## Acknowledgments

The authors are grateful to all the staff at the Centre de Recherche biomédicale En pathogènes Infectieux et Pathologies Associées (CREIPA) and the Unité Mixte de Recherche sur les Agents Infectieux et leur Pathologie (UMRAIP) for their invaluable support in participant recruitment in Bitam and Minvoul, and for their assistance with sample processing in both Libreville and, Bitam . Special thanks are also to Bifolossi Medical Center staff for hosting our study. We extend our gratitude to the village chiefs and other relevant authorities for their support throughout this research. Finally, we thank all the participants whose cooperation made this study possible.

## Author contributions

**Conceptualization:** Noé Patrick M'Bondoukwé, Denise Patricia Mawili Mboumba, Marielle Karine Bouyou Akotet.

**Data curation:** Luccheri Ndong Akomezoghe, Noé Patrick M'Bondoukwé.

**Formal analysis:** Luccheri Ndong Akomezoghe, Noé Patrick M'Bondoukwé.

**Funding acquisition:** Noé Patrick M'Bondoukwé, Marielle Karine Bouyou Akotet.

**Investigation:** Luccheri Ndong Akomezoghe, Noé Patrick M'Bondoukwé, Jacques Mari Ndong Ngomo, Bridy Chesly Moutombi Ditombi, Coella Joyce Mihindou, Roger Hadry Sibi Matotou, Valentin Migueba Migueba.

**Methodology:** Luccheri Ndong Akomezoghe, Noé Patrick M'Bondoukwé, Denise Patricia Mawili Mboumba.

**Project administration:** Denise Patricia Mawili Mboumba, Marielle Karine Bouyou Akotet.

**Resources:** Marielle Karine Bouyou Akotet.

**Supervision:** Denise Patricia Mawili Mboumba, Marielle Karine Bouyou Akotet.

**Validation:** Denise Patricia Mawili Mboumba, Marielle Karine Bouyou Akotet.

**Visualization:** Luccheri Ndong Akomezoghe.

**Writing – original draft:** Luccheri Ndong Akomezoghe, Noé Patrick M'Bondoukwé, Denise Patricia Mawili Mboumba.

**Writing – review & editing:** Luccheri Ndong Akomezoghe, Noé Patrick M'Bondoukwé, Denise Patricia Mawili Mboumba, Marielle Karine Bouyou Akotet.

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
