## [Decision Letter · Decision Letter 0]

Dear Dr M'Bondoukwé,

Thank you very much for submitting your manuscript "Efficacy and Safety of Albendazole 400 mg for 30 Days in Adult Patients with Low Loa loa Microfilaremia: A Non-Inferiority Randomized Controlled Trial Compared to Ivermectin" for consideration at PLOS Neglected Tropical Diseases. As with all papers reviewed by the journal, your manuscript was reviewed by members of the editorial board and by several independent reviewers. In light of the reviews (below this email), we would like to invite the resubmission of a significantly-revised version that takes into account the reviewers' comments. 

We cannot make any decision about publication until we have seen the revised manuscript and your response to the reviewers' comments. Your revised manuscript is also likely to be sent to reviewers for further evaluation.

Sincerely,

Feng Xue, Ph.D.

Guest Editor

Jong-Yil Chai

Section Editor

Reviewer's Responses to Questions

**Key Review Criteria Required for Acceptance?**

**Methods**

-Are the objectives of the study clearly articulated with a clear testable hypothesis stated?

-Is the study design appropriate to address the stated objectives?

-Is the population clearly described and appropriate for the hypothesis being tested?

-Is the sample size sufficient to ensure adequate power to address the hypothesis being tested?

-Were correct statistical analysis used to support conclusions?

-Are there concerns about ethical or regulatory requirements being met?

Reviewer #1: Just a comment regarding samples size, authors didn't mention how they calculate it

Reviewer #2: The objectives are clearly articulated

The study design was appropriate-

The study population are some how described

The sample size is adequate

There is no ethical concerns

Reviewer #3: All the. Methods are clearly described.

**Results**

-Does the analysis presented match the analysis plan?

-Are the results clearly and completely presented?

-Are the figures (Tables, Images) of sufficient quality for clarity?

Reviewer #1: yes

Reviewer #2: -Does the analysis presented match the analysis plan? : Yes

-Are the results clearly and completely presented?: somehow yes

-Are the figures (Tables, Images) of sufficient quality for clarity?: yes

Reviewer #3: The result are compiled in a good way for good understamding.

**Conclusions**

-Are the conclusions supported by the data presented?

-Are the limitations of analysis clearly described?

-Do the authors discuss how these data can be helpful to advance our understanding of the topic under study?

-Is public health relevance addressed?

Reviewer #1: yes

Reviewer #2: -Are the conclusions supported by the data presented?: Yes

-Are the limitations of analysis clearly described?: No

-Do the authors discuss how these data can be helpful to advance our understanding of the topic under study? Not really

-Is public health relevance addressed? Yes

Reviewer #3: Appropriate

**Editorial and Data Presentation Modifications?**

Reviewer #1: Authors need to improve the language they used as well as they should mention how they calculate the sample size and software of data analysis used to obtain the results

Reviewer #2: (No Response)

Reviewer #3: Ok

**Summary and General Comments**

Reviewer #1: N/A

Reviewer #2: The authors of this manuscript entitled “Efficacy and Safety of Albendazole 400 mg for 30 Days in Adult Patients with Loa loa Microfilaremia: A Non-Inferiority Randomized Controlled Trial Compared to Ivermectin “have reported an important NTD like issue. Indeed, up to date there is no clear cutting treatment against Loa loa, a vector born neglected diseases, causing enormously morbidity in endemic areas.

While the manuscript deserves to be published some minors reviews are needed and including:

Abstract:

• Authos should add in abstract section, the period (year(s)) where the study was conducted.?

• The technique(s) used for loa loa diagnostic?

• How long Albendazole was given before the 30 days followed up?

• Unfortunately the study was not designed for the prevalence, therefore I am wondering why it is so important to report this in the abstract as one of your main finding?

• What is the risk difference implied? Should this been proportion difference?

• The sample size is missing as well in the abstract section(how many participants were allocated to each group) the gender and age rans and so are missing .

• The abstract is missing the perspective for this important study.

Introduction

• Line 80 what this sentence “A significant body of research suggests that loiasis should be regarded as a major health issue” means

Material and Methods:

• Do you have a reference for the 20.2 % of Loa loa in the Northern?

• Are there any rational for chosen Loa loa <8000 and limited to below 3500?

• The reader may be confused, are the 10 microliter served for thick blood smear and the leucoconcentration technique? Any description of the last?

• The randomization procedure is not clear enough. How participant were selected to each arm is not clear.?

• Was the trial blinding or open? Please clarify, the choice here is subject of the consequence to the validation of the result.

Results:

• How were the 48 participants selected from 406 positive participants?

• What are the reason to exclude the 21 excluded participants?

• On the chart flow the 406 is not added up with 394 +48 please revise the numbers?

• What is the sequence of allocation of study participant?

• Author may structure better the results section. Having a heading of Efficacy of treatments on the reduction of microfilaremia and sub heading as PPT analysis and ITT analysis.

• It is not clear if the assessment of efficacy is post treatment. If so the assessment of those receiving Albendazole will occur 30 days after the 30 days course of treatment. Then 30 days after that of ivermectin

• Regarding safety, adverse events is vague, would author specify which events are we talking about, the new symptoms or worsen symptoms of existing one?

Discussion:

Line 264 and 287, we suggested to the author to avoid monthly treatment of albendazole and say “month course”

Reviewer #3: Overall paper is written in a good way

PLOS authors have the option to publish the peer review history of their article (what does this mean? ). If published, this will include your full peer review and any attached files.

**Do you want your identity to be public for this peer review?** For information about this choice, including consent withdrawal, please see our Privacy Policy .

Reviewer #1: No

Reviewer #2: No

Reviewer #3: No
---

## [Decision Letter · Decision Letter 1]

PNTD-D-24-01028R1Efficacy and Safety of Albendazole 400 mg for 30 Days Compared to Single Dose of Ivermectin in Adult Patients with Low Loa loa Microfilaremia: A Non-Inferiority Randomized Controlled TrialPLOS Neglected Tropical DiseasesDear Dr. M'Bondoukwé,

Thank you for submitting your manuscript to PLOS Neglected Tropical Diseases. After careful consideration, we feel that it has merit but does not fully meet PLOS Neglected Tropical Diseases's publication criteria as it currently stands. Therefore, we invite you to submit a revised version of the manuscript that addresses the points raised during the review process.

Please submit your revised manuscript within 30 days Dec 25 2024 11:59PM. If you will need more time than this to complete your revisions, please reply to this message or contact the journal office at plosntds@plos.org. Please include the following items when submitting your revised manuscript: * A rebuttal letter that responds to each point raised by the editor and reviewer(s). You should upload this letter as a separate file labeled 'Response to Reviewers '. This file does not need to include responses to any formatting updates and technical items listed in the 'Journal Requirements' section below. * A marked-up copy of your manuscript that highlights changes made to the original version. You should upload this as a separate file labeled 'Revised Manuscript with Track Changes '. * An unmarked version of your revised paper without tracked changes. You should upload this as a separate file labeled 'Manuscript '. If you would like to make changes to your financial disclosure, competing interests statement, or data availability statement, please make these updates within the submission form at the time of resubmission. Guidelines for resubmitting your figure files are available below the reviewer comments at the end of this letter. We look forward to receiving your revised manuscript. Kind regards, Feng Xue, Ph.D.Guest EditorPLOS Neglected Tropical DiseasesJong-Yil ChaiSection EditorPLOS Neglected Tropical Diseases

Shaden Kamhawi

co-Editor-in-Chief

Paul Brindley

co-Editor-in-Chief

**Journal Requirements:**

At this stage, the following Authors/Authors require contributions: Luccheri Ndong Akomezoghe, Noé Patrick M'Bondoukwé, Denise Patricia Mawili Mboumba, Jacques Mari Ndong Ngomo, Bridy Chesli Moutombi Ditombi, Coella Joyce Mihindou, Hadry Roger Sibi Matotou, Valentin Migueba Migueba, and Marielle Karine Bouyou Akotet. Please ensure that the full contributions of each author are acknowledged in the "Add/Edit/Remove Authors" section of our submission form.

2) Please amend your detailed Financial Disclosure statement. This is published with the article. It must therefore be completed in full sentences and contain the exact wording you wish to be published. Please ensure that the funders and grant numbers match between the Financial Disclosure field and the Funding Information tab in your submission form. Note that the funders must be provided in the same order in both places as well.

- State the initials, alongside each funding source, of each author to receive each grant. For example: "This work was supported by the National Institutes of Health (####### to AM; ###### to CJ) and the National Science Foundation (###### to AM).".

**Reviewers' comments:**

Reviewer's Responses to Questions

**Key Review Criteria Required for Acceptance?**

**Methods**

-Are the objectives of the study clearly articulated with a clear testable hypothesis stated?

-Is the study design appropriate to address the stated objectives?

-Is the population clearly described and appropriate for the hypothesis being tested?

-Is the sample size sufficient to ensure adequate power to address the hypothesis being tested?

-Were correct statistical analysis used to support conclusions?

-Are there concerns about ethical or regulatory requirements being met?

Reviewer #1: Yes

Reviewer #2: (No Response)

Reviewer #4: yes

**Results**

-Does the analysis presented match the analysis plan?

-Are the results clearly and completely presented?

-Are the figures (Tables, Images) of sufficient quality for clarity?

Reviewer #1: Yes

Reviewer #2: (No Response)

Reviewer #4: yes

**Conclusions**

-Are the conclusions supported by the data presented?

-Are the limitations of analysis clearly described?

-Do the authors discuss how these data can be helpful to advance our understanding of the topic under study?

-Is public health relevance addressed?

Reviewer #1: Yes

Reviewer #2: (No Response)

Reviewer #4: yes,

**Editorial and Data Presentation Modifications?**

Reviewer #1: Minor Revision

Reviewer #2: (No Response)

Reviewer #4: its ok

**Summary and General Comments**

Reviewer #1: English language needs to be improved. Moreover, research team should write more in Discussion sector

Reviewer #2: The authors respond well to my comments. However, some questions are remaining:

1- How many participants did have more than 3500 microfilariae?

2- Were other participants infected and not randomized been treated? If so, how?

3- How were the 48 participants selected from the 358 infected volunteers?

4- In introduction, the author justifies the advantage for using albendazole to treat Loa loa. Among those, there is a cost of albendazole which was 5$. It is not clear if the 5$ stand for the 30 days course of for one tablet. For the last it will cost 150$ vs 20$ and then Ivermectin seems in that case aTordable. The use of Albendazole instead of Ivermectin seems to be related to the safety rather to the cost. This need to be clarifies.

5- The non-inferiority finding was not tested. It will be interesting to see if the reduction rate between Albendazole and Ivermectin is not statistically significant.

Reviewer #4: This manuscript presents an interesting comparison of the efficacy and safety of Albendazole 400 mg for 30 days versus a single dose of Ivermectin in adult patients with Loa loa microfilaremia. Overall, the authors have successfully incorporated most of the reviewers' suggestions into the revised manuscript.

However, I recommend strengthening the discussion on the public health relevance and implications of the study. Specifically, the manuscript should address the potential challenges related to treatment adherence for a 30-day Albendazole regimen compared to the single-dose Ivermectin treatment, highlighting this as a limitation of the study. Additionally, further elaboration on how these findings can guide health surveillance in endemic regions would enhance the manuscript's impact.

PLOS authors have the option to publish the peer review history of their article (what does this mean? ). If published, this will include your full peer review and any attached files.

**Do you want your identity to be public for this peer review?** For information about this choice, including consent withdrawal, please see our Privacy Policy .

Reviewer #1: **Yes: ** Noor Kifah Al-Tameemi

Reviewer #2: No

Reviewer #4: **Yes: ** JOSÉ LUIZ FERNANDES VIEIRA

---

## [Decision Letter · Decision Letter 2]

PNTD-D-24-01028R2Efficacy and Safety of Albendazole 400 mg for 30 Days Compared to Single Dose of Ivermectin in Adult Patients with Low Loa loa Microfilaremia: A Non-Inferiority Randomized Controlled TrialPLOS Neglected Tropical Diseases Dear Dr. M'Bondoukwé, Thank you for submitting your manuscript to PLOS Neglected Tropical Diseases. After careful consideration, we feel that it has merit but does not fully meet PLOS Neglected Tropical Diseases's publication criteria as it currently stands. Therefore, we invite you to submit a revised version of the manuscript that addresses the points raised during the review process. Please submit your revised manuscript within 30 days Feb 15 2025 11:59PM. If you will need more time than this to complete your revisions, please reply to this message or contact the journal office at plosntds@plos.org. Please include the following items when submitting your revised manuscript: * A rebuttal letter that responds to each point raised by the editor and reviewer(s). You should upload this letter as a separate file labeled 'Response to Reviewers '. This file does not need to include responses to any formatting updates and technical items listed in the 'Journal Requirements' section below. * A marked-up copy of your manuscript that highlights changes made to the original version. You should upload this as a separate file labeled 'Revised Manuscript with Track Changes '. * An unmarked version of your revised paper without tracked changes. You should upload this as a separate file labeled 'Manuscript '. If you would like to make changes to your financial disclosure, competing interests statement, or data availability statement, please make these updates within the submission form at the time of resubmission. Guidelines for resubmitting your figure files are available below the reviewer comments at the end of this letter. We look forward to receiving your revised manuscript. Kind regards, Feng Xue, Ph.D.Guest EditorPLOS Neglected Tropical Diseases Jong-Yil ChaiSection EditorPLOS Neglected Tropical Diseases

Shaden Kamhawi

co-Editor-in-Chief

Paul Brindley

co-Editor-in-Chief

**Journal Requirements:**

At this stage, the following Authors/Authors require contributions: Luccheri Ndong Akomezoghe, Noé Patrick M'Bondoukwé, Denise Patricia Mawili Mboumba, Jacques Mari Ndong Ngomo, Bridy Chesli Moutombi Ditombi, Coella Joyce Mihindou, Hadry Roger Sibi Matotou, Valentin Migueba Migueba, and Marielle Karine Bouyou Akotet. Please ensure that the full contributions of each author are acknowledged in the "Add/Edit/Remove Authors" section of our submission form.

2) We have noticed that you have uploaded Supporting Information files, but you have not included a list of legends. Please add a full list of legends for your Supporting Information files after the references list.

3) Please amend your detailed Financial Disclosure statement. This is published with the article. It must therefore be completed in full sentences and contain the exact wording you wish to be published. State what role the funders took in the study. If the funders had no role in your study, please state: "The funders had no role in study design, data collection and analysis, decision to publish, or preparation of the manuscript.".

**Reviewers' comments:** Reviewer's Responses to Questions

**Key Review Criteria Required for Acceptance?**

**Methods**

-Are the objectives of the study clearly articulated with a clear testable hypothesis stated?

-Is the study design appropriate to address the stated objectives?

-Is the population clearly described and appropriate for the hypothesis being tested?

-Is the sample size sufficient to ensure adequate power to address the hypothesis being tested?

-Were correct statistical analysis used to support conclusions?

-Are there concerns about ethical or regulatory requirements being met?

Reviewer #1: Yes

Reviewer #2: ok

Reviewer #4: Among these questions I recommend these correction in statistical analysis:Comparisons among different time points should specify the use of analysis of variance or appropriate non-parametric tests, as applicable.

The objective of the study are clearly defined. The sample size is adequate. There is no concern about ethical requirement. The study design is appropriate.

**Results**

-Does the analysis presented match the analysis plan?

-Are the results clearly and completely presented?

-Are the figures (Tables, Images) of sufficient quality for clarity?

Reviewer #1: Yes

Reviewer #2: ok

Reviewer #4: The second line of Table 1 appears to be a footnote. This formatting should be corrected for consistency and clarity.

The analysis match the proposed plain

The authors need to clearly differentiate between adverse events caused by the drug and those attributable to the disease itself

**Conclusions**

-Are the conclusions supported by the data presented?

-Are the limitations of analysis clearly described?

-Do the authors discuss how these data can be helpful to advance our understanding of the topic under study?

-Is public health relevance addressed?

Reviewer #1: Yes

Reviewer #2: ok

Reviewer #4: The sentence, “The prevalence of Loa loa infection is notably high in the investigated villages of the Woleu-Ntem province Gabon,” is not directly relevant to the study’s primary findings and should be removed from the conclusion.

**Editorial and Data Presentation Modifications?**

Reviewer #1: Accept

Reviewer #2: ok

Reviewer #4: The authors need to clearly differentiate between adverse events caused by the drug and those attributable to the disease itself.

Considering that the authors have patient weight data, why was the albendazole (ALB) dose not calculated based on weight, as was described for ivermectin (IVM)? This could provide a more precise understanding of dosage effects.The title, “Adverse events and symptoms during treatment”, is ambiguous and could lead to misunderstandings. Consider rephrasing it to be more precise.

I find it difficult to classify “increased appetite in 20.0% (n=2/10)” as a symptom or an adverse event. This categorization should be revisited.

**Summary and General Comments**

Reviewer #1: The authors responded well to reviewers comments

Reviewer #2: ok

Reviewer #4: This is an interesting manuscript for an endemic area. Some modifications, cited above, are required for a better understanding of readers.

PLOS authors have the option to publish the peer review history of their article (what does this mean? ). If published, this will include your full peer review and any attached files.

**Do you want your identity to be public for this peer review?** For information about this choice, including consent withdrawal, please see our Privacy Policy .

Reviewer #1: **Yes: ** Noor Kifah Al-Tameemi

Reviewer #2: No

Reviewer #4: No

**Figure resubmission:** While revising your submission, please upload your figure files to the Preflight Analysis and Conversion Engine (PACE) digital diagnostic tool, https://pacev2.apexcovantage.com/. PACE helps ensure that figures meet PLOS requirements. To use PACE, you must first register as a user. Registration is free. Then, login and navigate to the UPLOAD tab, where you will find detailed instructions on how to use the tool. If you encounter any issues or have any questions when using PACE, please email PLOS at figures@plos.org. Please note that Supporting Information files do not need this step. If there are other versions of figure files still present in your submission file inventory at resubmission, please replace them with the PACE-processed versions.**Reproducibility:** To enhance the reproducibility of your results, we recommend that authors of applicable studies deposit laboratory protocols in protocols.io, where a protocol can be assigned its own identifier (DOI) such that it can be cited independently in the future. Additionally, PLOS ONE offers an option to publish peer-reviewed clinical study protocols. Read more information on sharing protocols at https://plos.org/protocols?utm_medium=editorial-email&utm_source=authorletters&utm_campaign=protocols

---

## [Decision Letter · Decision Letter 3]

PNTD-D-24-01028R3Efficacy and Safety of Albendazole 400 mg for 30 Days Compared to Single Dose of Ivermectin in Adult Patients with Low Loa loa Microfilaremia: A Non-Inferiority Randomized Controlled TrialPLOS Neglected Tropical DiseasesDear Dr. M'Bondoukwé, Thank you for submitting your manuscript to PLOS Neglected Tropical Diseases. After careful consideration, we feel that it has merit but does not fully meet PLOS Neglected Tropical Diseases's publication criteria as it currently stands. Therefore, we invite you to submit a revised version of the manuscript that addresses the points raised during the review process. Please submit your revised manuscript within 30 days Apr 09 2025 11:59PM. If you will need more time than this to complete your revisions, please reply to this message or contact the journal office at plosntds@plos.org. Please include the following items when submitting your revised manuscript: * A rebuttal letter that responds to each point raised by the editor and reviewer(s). You should upload this letter as a separate file labeled 'Response to Reviewers '. This file does not need to include responses to any formatting updates and technical items listed in the 'Journal Requirements' section below. * A marked-up copy of your manuscript that highlights changes made to the original version. You should upload this as a separate file labeled 'Revised Manuscript with Track Changes '. * An unmarked version of your revised paper without tracked changes. You should upload this as a separate file labeled 'Manuscript '.

We look forward to receiving your revised manuscript. Kind regards, Feng Xue, Ph.D.Guest EditorPLOS Neglected Tropical Diseases Jong-Yil ChaiSection EditorPLOS Neglected Tropical Diseases

Shaden Kamhawi

co-Editor-in-Chief

Paul Brindley

co-Editor-in-Chief

**Journal Requirements:**

At this stage, the following Authors/Authors require contributions: Luccheri Ndong Akomezoghe, Noé Patrick M'Bondoukwé, Denise Patricia Mawili Mboumba, Jacques Mari Ndong Ngomo, Bridy Chesly Moutombi Ditombi, Coella Joyce Mihindou, Roger Hadry Sibi Matotou, Valentin Migueba Migueba, and Marielle Karine Bouyou Akotet. Please ensure that the full contributions of each author are acknowledged in the "Add/Edit/Remove Authors" section of our submission form.

2) We have noticed that you have uploaded Supporting Information files, but you have not included a list of legends. Please add a full list of legends for your Supporting Information files after the references list.

3) We note that your Data Availability Statement is currently as follows: "All data are available without restriction and data set was shared with the manuscript to replicate obtained results.All data underlying the findings described are within the manuscript itself.All data are in the manuscript and/or supporting information files". Please confirm at this time whether or not your submission contains all raw data required to replicate the results of your study. Authors must share the “minimal data set” for their submission. PLOS defines the minimal data set to consist of the data required to replicate all study findings reported in the article, as well as related metadata and methods (https://journals.plos.org/plosone/s/data-availability#loc-minimal-data-set-definition).

- The points extracted from images for analysis..

4) Please ensure that the funders and grant numbers match between the Financial Disclosure field and the Funding Information tab in your submission form. Note that the funders must be provided in the same order in both places as well. State the initials, alongside each funding source, of each author to receive each grant. For example: "This work was supported by the National Institutes of Health (####### to AM; ###### to CJ) and the National Science Foundation (###### to AM).".

**Reviewers' comments:** Reviewer's Responses to Questions

**Key Review Criteria Required for Acceptance?**

**Methods:**

-Are the objectives of the study clearly articulated with a clear testable hypothesis stated?

-Is the study design appropriate to address the stated objectives?

-Is the population clearly described and appropriate for the hypothesis being tested?

-Is the sample size sufficient to ensure adequate power to address the hypothesis being tested?

-Were correct statistical analysis used to support conclusions?

-Are there concerns about ethical or regulatory requirements being met?

Reviewer #1: Yes

Reviewer #2: yes

Reviewer #4: The objectives of the study are clearly articulated with a clear testable hypothesis

The design of the study is appropriate

The population is clearly described in the study

The sample size is ok

The ethical statement is ok

the statistical analysis is adequate

**Results:**

-Does the analysis presented match the analysis plan?

-Are the results clearly and completely presented?

-Are the figures (Tables, Images) of sufficient quality for clarity?

Reviewer #1: Yes

Reviewer #2: yes

Reviewer #4: (No Response)

**Conclusions:**

-Are the conclusions supported by the data presented?

-Are the limitations of analysis clearly described?

-Do the authors discuss how these data can be helpful to advance our understanding of the topic under study?

-Is public health relevance addressed?

Reviewer #1: Yes

Reviewer #2: yes

Reviewer #4: yes the conclusion supports the data presented

yes the study limitations were clarified

yes

the study is relevant for health

**Editorial and Data Presentation Modifications?**

Reviewer #1: English language need to be enhanced

Reviewer #2: (No Response)

Reviewer #4: there is no data to be revised, but I believed that improved appetite is not an adverse reaction

**Summary and General Comments:**

Reviewer #1: It's good paper but English language need to be enhanced

Reviewer #2: I am still not convinced with the cost of Albendazole. So to aleviate any doubt and to facilitate public health implementation of this important findings, author should provide the reference including the vendor at national levle as wellas the manufacturer contacts. Should a patient got 50 tablets of Albendazole at the 5$ in Gabon?

Reviewer #4: The revised version follow the scientific criteria for publication in PLOS Neglected Tropical Diseases. There are minor grammatical errors. Additionally I believed that improved appetite is not a adverse reaction

PLOS authors have the option to publish the peer review history of their article (what does this mean? ). If published, this will include your full peer review and any attached files.

**Do you want your identity to be public for this peer review?** For information about this choice, including consent withdrawal, please see our Privacy Policy .

Reviewer #1: **Yes: ** Noor Kifah Al-Tameemi

Reviewer #2: No

Reviewer #4: **Yes: ** José Luiz Fernandes Vieira

**Figure resubmission:**

While revising your submission, please upload your figure files to the Preflight Analysis and Conversion Engine (PACE) digital diagnostic tool, https://pacev2.apexcovantage.com/. PACE helps ensure that figures meet PLOS requirements. To use PACE, you must first register as a user. Registration is free. Then, login and navigate to the UPLOAD tab, where you will find detailed instructions on how to use the tool. If you encounter any issues or have any questions when using PACE, please email PLOS at figures@plos.org. Please note that Supporting Information files do not need this step. If there are other versions of figure files still present in your submission file inventory at resubmission, please replace them with the PACE-processed versions. **Reproducibility:** To enhance the reproducibility of your results, we recommend that authors of applicable studies deposit laboratory protocols in protocols.io, where a protocol can be assigned its own identifier (DOI) such that it can be cited independently in the future. Additionally, PLOS ONE offers an option to publish peer-reviewed clinical study protocols. Read more information on sharing protocols at https://plos.org/protocols?utm_medium=editorial-email&utm_source=authorletters&utm_campaign=protocols

---

## [Decision Letter · Decision Letter 4]

Dear Dr M'Bondoukwé,

We are pleased to inform you that your manuscript 'Efficacy and Safety of Albendazole 400 mg for 30 Days Compared to Single Dose of Ivermectin in Adult Patients with Low Loa loa Microfilaremia: A Non-Inferiority Randomized Controlled Trial' has been provisionally accepted for publication in PLOS Neglected Tropical Diseases.

Best regards,

Feng Xue, Ph.D.

Guest Editor

Jong-Yil Chai

Section Editor

Shaden Kamhawi

co-Editor-in-Chief

Paul Brindley

co-Editor-in-Chief

Reviewer's Responses to Questions

**Key Review Criteria Required for Acceptance?**

**Methods**

-Are the objectives of the study clearly articulated with a clear testable hypothesis stated?

-Is the study design appropriate to address the stated objectives?

-Is the population clearly described and appropriate for the hypothesis being tested?

-Is the sample size sufficient to ensure adequate power to address the hypothesis being tested?

-Were correct statistical analysis used to support conclusions?

-Are there concerns about ethical or regulatory requirements being met?

Reviewer #2: no comment

Reviewer #4: Yes, the objetives of th Study are articulares with a testable hypothesis.

The Study design is adequate to objectives

The population is cleary defined

Although the sample size was small, the authors provided a justification in the study’s limitations section.

The authors forgot to include the significance level in their hyphotesis, but the statistical tests used are adequated

The ethical issue mas clearly defined by the authors.

Reviewer #5: (No Response)

Reviewer #6: yes

Reviewer #7: - See attached redlined manuscript for suggested minor edits.

- "Exclusion criteria" should refer to determination of eligibility prior to enrollment. I recommend changing the authors' use of the term "exclusion criteria" to "withdrawal criteria" since they are referring to post-enrollment withdrawal of participants from the study.

- The authors claim that the study was blinded, yet one group received one dose of medication whereas the other received 30 days of daily medication. How could that have been blinded?

**Results**

-Does the analysis presented match the analysis plan?

-Are the results clearly and completely presented?

-Are the figures (Tables, Images) of sufficient quality for clarity?

Reviewer #2: no comment

Reviewer #4: yes

yes

yes

Reviewer #5: (No Response)

Reviewer #6: yes

Reviewer #7: - See attached redlined manuscript for suggested minor edits.

- Please add error bars/95% CIs to the figures, if possible.

- In the "Incidence and evolution of adverse events during treatment" section, please state the overall percentage of participant experiencing AEs, not just the percentage of those experiencing AEs.

**Conclusions**

-Are the conclusions supported by the data presented?

-Are the limitations of analysis clearly described?

-Do the authors discuss how these data can be helpful to advance our understanding of the topic under study?

-Is public health relevance addressed?

Reviewer #2: no comment

Reviewer #4: yes

yes

yes

yes

Reviewer #5: (No Response)

Reviewer #6: yes

Reviewer #7: - See attached redlined manuscript for suggested minor edits.

**Editorial and Data Presentation Modifications?**

Reviewer #2: No further modification is required

Reviewer #4: Insert the statistical signifcance level

Adjust Republic of Congo in the discusion section

Reviewer #5: (No Response)

Reviewer #6: minor revision

Reviewer #7: - See attached redlined manuscript for suggested minor edits.

**Summary and General Comments**

Reviewer #2: The manuscript deserved to be published

Reviewer #4: no

Reviewer #5: (No Response)

Reviewer #6: This clinical trial evaluated daily albendazole versus single dose ivermectin in reducing microfilaremia in loiasis patients in Gabon. The main findings are an around 80% and 90% reduction in mf counts at day 30. The authors have to be congratulated for the conduct of this clinical trial as we lack well performed studies for loiasis.

The study was diligently planned, conducted and analysed. Overall the sample size is limited but large enough to have reliable estimates for the efficacy of the employed treatments. I would argue that the sample size was too low to obtain conclusive informations about the safety of the drug regimens. Safety findings are often rare and therefore the overall limited sample size did not allow for assessment of rare adverse events potentially linked to the treatment regimens.

Finally the relatively short observation period of 30 days allows to well characterize the initial response to the treatment but does not allow to obtain information about the long-term results. This could be added to the limitations.

Reviewer #7: - See attached redlined manuscript for suggested minor edits.

PLOS authors have the option to publish the peer review history of their article (what does this mean? ). If published, this will include your full peer review and any attached files.

**Do you want your identity to be public for this peer review?** For information about this choice, including consent withdrawal, please see our Privacy Policy .

Reviewer #2: No

Reviewer #4: **Yes: ** JOSÉ LUIZ FERNANDES VIEIRA

Reviewer #5: No

Reviewer #6: No

Reviewer #7: No

---

## [Editor Report · Acceptance letter]

Dear Dr M'Bondoukwé,

We are delighted to inform you that your manuscript, "Efficacy and Safety of Albendazole 400 mg for 30 Days Compared to Single Dose of Ivermectin in Adult Patients with Low Loa loa Microfilaremia: A Non-Inferiority Randomized Controlled Trial," has been formally accepted for publication in PLOS Neglected Tropical Diseases.

Best regards,

Shaden Kamhawi

co-Editor-in-Chief

Paul Brindley

co-Editor-in-Chief
